**Comparative Analysis of $\mu\,(I)$ and Voellmy-Type Grain Flow Rheologies in**
**Geophysical Mass Flows: Insights from Theoretical and Real Case Studies**
Yu Zhuang[1,2,3,4], Brian W. McArdell[5], Perry Bartelt[3,4]
[1]College of Civil Engineering, Hunan University, Changsha 410082, China
[2]Key Laboratory of Building Safety and Energy Efficiency of the Ministry of Education, Changsha
410082, China
[3]WSL Institute for Snow and Avalanche Research SLF, Davos Dorf 7260, Switzerland
[4]Climate Change, Extremes and Natural Hazards in Alpine Regions Research Centre CERC
[5]Swiss Federal Institute for Forest, Snow and Landscape Research WSL, Birmensdorf 8903, Switzerland
Corresponding author: Yu Zhuang (zhuangyu@hnu.edu.cn)
**Abstract**
The experimental-based $\mu(I)$ rheology is now prevalent to describe the movement of gravitational mass
flows. We reinterpret the $\mu(I)$ rheology as a Voellmy-type relationship to highlight its connection to
grain flow theory and demonstrate its practical applications. Using one-dimensional block modeling and
two real-world case studies—the 2017 Piz Cengalo rock-ice avalanche and an experimental snow
avalanche at the Swiss Vallée de la Sionne test site—we demonstrate the relationship between the
dimensionless number $I$ and the granular temperature $R$, establishing the equivalence between $\mu(I)$
and widely-used Voellmy-type grain flow rheologies $\mu(R)$. Results indicate that $\mu(I)$ rheology utilizes
the dimensionless inertial number $I$ to mimic contributions of granular temperature/fluctuation energy
to flow behaviour. In terms of Voellmy, the $\mu(I)$ rheolgy contains a velocity-dependent turbulent
friction coefficient modelling shear thinning behavior. This turbulent friction assumes the production and
decay of fluctuation energy are in balance, exhibiting no difference during accelerative and depositional
phases of avalanche flow. The constant Coulomb friction coefficient prevents $\mu(I)$ rheology from
accurately modeling the dispositional characteristics of actual mass flows. The modeled evolution of the
snow avalanche using the $\mu(I)$ rheology is too slow, lagging 5 seconds behind the measured values.
More importantly, the calculated runout extends approximately 200 meters beyond the observed limits,
with significant deposit anomalies in the valley. By incorporating a non-steady production and decay of
fluctuation energy in the $\mu(R)$ framework, it becomes possible to achieve a good match with both the
measured velocities and the observed runout. Our results highlight the strengths and limitations of both
$\mu(I)$ and Voellmy $\mu(R)$ rheologies, bolstering the theoretical foundation of mass flow modeling while
revealing practical engineering challenges.
**Keywords:** $\mu(I)$ rheology; Voellmy-type Grain flow rheologies; Geophysical mass flows; Avalanche
risk assessment
**1. Introduction**
Creating dependable methods to forecast the runout and deposition characteristics of geophysical mass
flows stands as a fundamental challenge in natural hazard research. Long runout mass flows, like debris
flows, rock/ice avalanches and snow slides, occur in complex mountain terrain and exhibit an array of
complex outcomes depending on their initial material composition and dynamic interactions with the
substrate. These mass movements of granular composition exhibit significant mobility, vast energy, and
diverse flow patterns, posing challenges for prediction using numerical models (Crosta et al., 2007;
Hürlimann et al., 2015; Iverson et al., 2015; Frigo et al., 2021; Shugar et al., 2021). A crucial element for
precise modeling of their various behaviors is the development of a universal rheology capable of
accurately capturing their granular motion, including long-distance travel, transitions between flow
regimes, and eventual deposition.

45   Presently, two primary types of numerical models dominate in engineering practice: discrete

46 element methodologies (Scaringi et al., 2018; Zhao & Crosta, 2018) and continuum approaches, often

47 employing depth-averaged techniques (Hungr & McDougall, 2009; Christen et al., 2010). Discrete

48 approaches simulate particle interactions, incorporating fragmentation processes, thus adeptly portraying

49 the complex behavior of flowing granular materials (Katz et al., 2014; Zhao et al., 2017; Zhuang et al.,

50 2023a). Nonetheless, accurately replicating the sheer volume of particles within real geophysical mass

51 flows remains a formidable challenge, constraining their utility for solving large-scale problems due to

52 computational constraints. Conversely, the continuum approaches treat the mass flow as a "granular fluid"

53 consisting of particle ensembles. They utilize a series of differential equations to calculate the flow

54 process, offering high computational efficiency (McDougall & Hungr, 2004; Christen et al., 2010;

55 Mergili et al., 2017). Because existing continuum approaches account for the essential process of ground

56 entrainment (Sovilla & Bartelt, 2002, Bartelt et al., 2018a), frictional heating and phase changes (Valero

57 et al., 2015; Bartelt et al., 2018b), they are somewhat more advanced than discrete element approaches

58 and thus have been widely used to assess mass flow hazard.

59   The Voellmy rheology (Voellmy, 1995) has a long tradition in the hazard mitigation community and

60 is applied to predict the velocity and runout of avalanches and debris flows (Hungr, 1995; Schraml et al.,

61 2015; Aaron et al., 2019; Zhuang et al., 2020). It defines the relationship $\mu(V)=S/N$ as follows:

62 $$\mu(V) = \frac{S}{N} = \mu_s + \frac{v^2}{\xi_0 h} \tag{1}$$

63 where $\mu_s$ considers the Coulomb friction at "stopping", $v$ is the flowing velocity, $\xi_0$ the "turbulent"

64 friction parameter; $h$ the flowing height. Voellmy considers $\mu_s$ to describe the "solid" behavior of the

65 flowing mass, whereas $\xi_0$ represents the "fluid"-like behavior. Because the Voellmy model is grounded

66 in clear physical principles and involves only two parameters, it is frequently used in hazard mitigation.

However, a major issue with the Voellmy model is that the travel resistance of mass flows varies
significantly with the flow regime (Gruber and Bartelt, 1998). In the Voellmy model, each flow regime
requires a distinct set of calibrated flow parameters; there is no universal parameter set available,
rendering the Voellmy approach somewhat makeshift. To address this issue, multiple researchers have
suggested incorporating the concept of granular temperature (fluctuation energy $R$) to accurately model
the flow of granular materials across both dense and fluidized flow regimes (Haff, 1983; Jenkins &
Savage, 1983; Jenkins & Mancini, 1987; Gubler, 1987; Buser & Bartelt, 2009). The term granular
temperature (fluctuation energy $R$) originates from thermodynamics and represents the kinetic energy
associated with random particle motions in the granular ensemble; it is defined based on the velocity
fluctuations of individual grains (Campbell, 2006). This approach involves adding an extra differential
equation to account for the generation and dissipation of kinetic energy due to random particle
movements (Bartelt et al., 2006). The fluctuation energy arises from shear-work rate $\dot{W}_f$ and decays by
dissipative granular interactions (Haff, 1983):

$$\frac{dR(t)}{dt} = \alpha \dot{W}_f(t) - \beta(R)R(t) \tag{2}$$

where $\alpha$ governs the production and $\beta$ governs the decay of the fluctuation energy. It is possible to
express the friction parameters ($\mu_s$, $\xi$) as a function of the fluctuation energy, named $\mu(R)$ rheology.
Within the Voellmy framework, the $\mu(R)$ rheology has the form (Christen et al., 2010; Zhuang et al.,

84    2024):

$$\mu(R) = \mu_s(R) + \frac{v^2}{\xi(R)h} \tag{3}$$

where $\mu_s(R) = \mu_s e^{-\frac{R(t)}{R_0}}$, $\xi(R) = \xi_0 e^{\frac{R(t)}{R_0}}$, the parameter $R_0$ scales the fluctuation energy. This $\mu(R)$
rheology has the advantage of modeling shear-thinning in avalanche flows, showing a better agreement
with observed front velocities and mapped deposition patterns of avalanches than the classic Voellmy
approach (Preuth et al., 2010; Bartelt et al., 2012).
Recently, the $\mu(I)$ rheology is newly proposed to describe the motion of geophysical flows. It arose
directly from the study of small-scale granular experiments (GDR MIDI, 2004; Jop et al., 2006):

$$\mu(I) = \frac{S}{N} = \mu_s + \frac{(\mu_2 - \mu_s)}{\frac{I_0}{I_n} + 1} \tag{4}$$

Similar to Voellmy, the model consists of two parts. The first part consists of the stopping friction $\mu_s$.
The second term is controlled by the inertial number $I_n$, which describes the ratio of inertial forces of
grains to imposed forces, and is defined as (GDR MIDI, 2004):

$$I_n = \frac{5}{2h} \frac{vd}{\sqrt{g_z h}} \tag{5}$$

where $d$ is the granule diameter and $g_z$ the slope perpendicular component of gravity. The model
contains two additional constant parameters, $I_0$ and $\mu_2$, which can be considered the friction at large
$I_n$. Because of its well-established experimental foundation, the $\mu(I)$ model has become popular in the
granular mechanics community and is applied in hazard practice (e.g., Longo et al., 2019; Liu et al.,
2022). Although there is broad interest and advocacy for its use, the physical implications of the $\mu(I)$
rheology are not completely understood, which restricts its widespread adoption.
In this study, we reformulate the $\mu(I)$ rheology as a Voellmy-type relationship. Through one-
dimensional block modeling, we investigate the equivalence and difference between the $\mu(I)$ and
Voellmy-type grain flow rheologies. Two historical cases—the 2017 Piz Cengalo rock-ice avalanche and
a snow avalanche at the Vallée de la Sionne test site in Switzerland—are further analyzed to demonstrate
the performance of the $\mu(I)$ rheology. The primary objective of this study is to establish the $\mu(I)$
rheology on a more robust theoretical framework, critically enhancing our understanding of its utility in
predicting the dynamics of geophysical mass flows. This endeavor is essential to establish a comparative
understanding of different models presently used in natural hazards practice.

## 2. Method and Data

### 2.1 Reformulation of the $\mu(I)$ rheology

The rheological model describes the relationship between the shear stress $S$ to the normal stress $N$ of the flowing mass. The comparison between the $\mu(V)$ and $\mu(I)$ rheologies is for practical applications intuitively made in $S$ vs $N$ space. Here, we vary the flow height (normal stress) and fix the velocity at a specific value to make the comparison, as presented in Fig. 1a. The quantitative and qualitative similarity between the $\mu(V)$ and $\mu(I)$ approaches in $S$ vs $N$ space suggests a mathematical relationship between the two models. In light of this, we have reformulated the $\mu(I)$ rheology using a Voellmy sum:

$$\mu(I) = \mu_s + \frac{v^2}{\xi(I)h} \tag{6}$$

where $\xi(I)$ characterizes the "turbulent friction" of the $\mu(I)$ model. We find:

$$\xi(I) = \frac{v\left[2\,I_0 h\sqrt{g_z h} + 5vd\right]}{5(\mu_2 - \mu_s)d} \tag{7}$$

Different from the constant $\xi_0$ value in the Voellmy, $\xi(I)$ is changing during the flowing process, and is dependent on the flowing velocity and height (Fig. 1b).

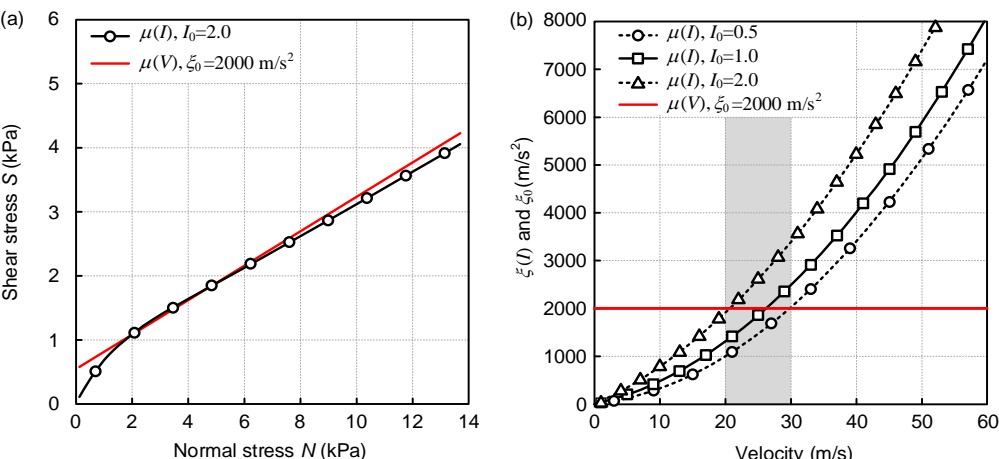

**Figure 1.** $\mu(I)$ vs $\mu(V)$ rheology for typical snow avalanche conditions, $v=20$ m/s and $\rho=300$ kg/m$^3$. For this example, we take $\mu_s=0.2679=\tan(15°)$ and $\mu_2=0.8391=\tan(40°)$. (a) The curve $I_0=2.0$ plotted against $\mu(V)$ with $\xi_0=2000$ m/s$^2$. Note the strong similarity between the $\mu(I)$ and $\mu(V)$ approaches in $S$ vs $N$ space. (b) Comparison of the $\mu(I)$ vs $\mu(V)$ rheologies in velocity space. $\xi(I)$ increases with velocity; $\xi(V)=\xi_0$ is constant. In the shaded region 20m/s $\leq v \leq$ 30m/s, the $\xi(I)$ and $\xi(V)$ values are similar.

## 2.2 One-dimensional block modeling analysis

The turbulent friction coefficient $\xi(I)$ is velocity-dependent. According to Fig. 1, the primary reason for the similarity of the two results is the selected velocity for the comparison $v$=20 m/s. For velocities outside this range, the $\xi(I)$ and $\xi(V)=\xi_0$=constant values differ (Fig. 1b). Therefore, to investigate the difference between $\mu(I)$ and $\mu(V,R)$, we must study the models over a wide range of velocities typical for a specific geophysical flow from initiation to runout.

For this purpose, we construct a one-dimensional block model. A block of height $h$ and mass $m$ starts from rest on a steep slope of 35° (release zone). After 30 s the block enters a transition zone of 20°, where it begins to decelerate. After 90 s the block enters a flat runout zone and stops. We calculate the speed and location of the block's center-of-mass; friction is given by $\mu(I)$, $\mu(V)$ and $\mu(R)$. The governing ordinary differential equations for this model are:

$$\frac{\mathrm{d}x(t)}{\mathrm{d}t} = v(t) \tag{8}$$

$$\frac{\mathrm{d}v(t)}{\mathrm{d}t} = g_x(t) - \mu(I,V,R)g_z(t) \tag{9}$$

where $x(t)$ is the flowing distance, $v(t)$ is the flowing velocity, and $(g_x,\ g_z)$ are the components of gravity acceleration.

We consider the motion of the center-of-mass to represent the motion of a granular, geophysical flow. Such simple, one-dimensional sliding block models of avalanche flow have been used extensively to calculate hazard maps (Perla et al., 1980). This approach allows us to compare the $\mu(I)$ and $\mu(V,R)$ rheologies in velocity space.

## 2.3 Case study of historical avalanches

According to the reformulation of the $\mu(I)$ rheology, $\xi(I)$ parameter is a function of both flowing height and velocity (Eq. 7), which is heavily dependent on the flowing regime and entrainment process. The

one-dimensional block model ignores the above essential features and processes. Therefore, we conduct
an analysis of two historical avalanche cases: 2017 Piz Cengalo rock-ice avalanche (Mergili et al., 2020)
and a snow avalanche (No. #20163017) that occurred in Vallée de la Sionne test site, Switzerland (Sovilla
et al., 2018). The Piz Cengalo avalanche occurred on 23th August, 2017 with a released rock volume of
~$3 \times 10^6$ m$^3$. The sliding mass entrained the glacial of $6 \times 10^5$ m$^3$ and formed a rock-ice avalanche. This
avalanche is well documented with laser scans of release and deposits, providing natural materials to
confirm the numerical model (Mergili et al., 2020; Walter et al., 2020). The snow avalanche (#20163017)
was artificially triggered on 18$^{th}$ January, 2016. The avalanche involved an initial volume of 86560 m$^3$
and a runout of ~2500 m. The difference between DEMs before and after the event indicated the deposit
structure, and cameras recorded the evolution of the snow avalanche. Detailed information about this
particular snow avalanche is presented in Sovilla et al., (2018).
We implement the Voellmy $\mu(V)$, $\mu(I)$ and $\mu(R)$ rheologies into a continuum approach-based
model RAMMS (Christen et al., 2010; Bartelt et al., 2018b; Zhuang et al., 2024) to elucidate the
performance and limitations of the $\mu(I)$ rheology in calculating the evolution of geophysical mass flows.
Detailed information about the well-established **RAMMS** model can be found in Christen et al. (2010),
Bartelt et al. (2016, 2018b), and Zhuang et al. (2024).
**3. Results**
**3.1 Rheology comparison using the one-dimensional block model**
**(1) The $\mu(I)$ and $\mu(V)$ rheologies in velocity space**
The direct comparison of $\mu(I)$ and $\mu(V)$ reveals that both models can produce similar runout (Fig. 2a),
and velocity (Fig. 2b). However, the $\mu(V)$ approach reaches a smaller peak velocity at the end of the
release zone but decelerates less strongly in the transition zone (Fig. 2b). In the end, the velocity at the
beginning of the runout zone is higher. This result can also be visualized in the depiction of location
through time (Fig. 2a). The Voellmy flow reaches the same runout distance but lags the $\mu(I)$ model
along the intermediate transition segment. Of interest is a direct comparison of $\mu(I)$ and $\mu(V)$ through
time (Fig. 2c). The $\mu(V)$ with constant $\xi_0$ reaches larger values (lower velocities) but decreases rapidly
during the transition to the flatter 20° slope, falling to values smaller than $\mu(I)$. Both models predict the
same $\mu$ values as the block enters the flat runout zone. According to Eq. 7, $\xi(I)$ increases with the
flowing velocity, indicating a shear-thinning type of behavior and therefore a smaller resistance in the
acceleration stage. The general model behavior over the three slope segments can be explained by the
fact that the constant $\xi_0$ value characterizes a mean value within the domain of possible $\xi(I)$ values.
Model parameters can be selected such that similar results are obtained; experiments are required to
determine which accelerative/decelerative behavior represents the best fit to observations. However,
there is a method to bring the two model approaches into equivalence.

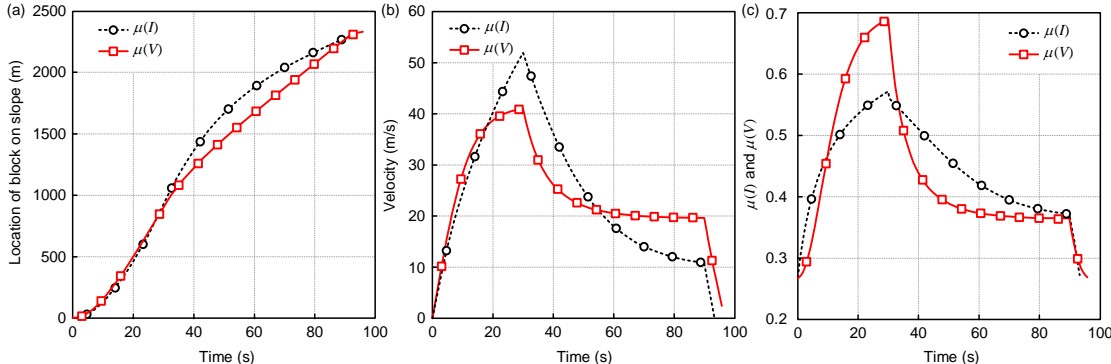

**Figure 2.** The $\mu(I)$ vs $\mu(V)$ rheologies in velocity space. (a) Location of center-of-mass over time. In
the transition zone the Voellmy model with constant $\xi_0$ lags the $\mu(I)$ model. (b) Velocity over time.
With a constant $\xi_0$ the Voellmy model tends to a steady velocity, albeit a lower velocity than $\mu(I)$. At
the end of the transition zone, the Voellmy model predicts a higher (steady state) velocity. (c) $S/N$ for
$\mu(I)$ and $\mu(V)$. The Voellmy model predicts higher friction before entering the transition zone.
**(2) The Voellmy grain-flow equivalent to $\mu(I)$: The $\mu(R)$ grain flow rheology**
The Voellmy-type $\mu(R)$ rheology is a function of granular temperature/fluctuation energy, which arises
from shearing work and decays by dissipative granular interactions. To better compare the $\mu(I)$ and
$\mu(R)$ rheologies, we made the Coulomb friction parameter $\mu_s(R)$ a constant but turbulent friction
parameter $\xi(R)$ a function of fluctuation energy, so that the two rheologies are in the same Voellmy-
type. When we re-solve the ordinary differential equations (Eqs. 8 and 9) with the additional production-
decay equation (Eq. 2) and the parameters $\alpha$ =0.05, $\beta$ =0.95, $\xi_0$ =500 m/s$^2$ and $R_0$ =6 kJ, we find a
remarkable duplication of the $\mu(I)$ results, with regard to the calculated location (Fig. 3a), velocity (Fig.
3b) and calculated $\mu(I)$ and $\mu(R)$ (Fig. 3c). In this comparison the $\mu(I)$ model employed the
following parameters, $I_0$ =1.0, $d$ =0.07 m, $\mu_2$ =tan(40$^\circ$) and $\mu_s$ =tan(15$^\circ$).

These results suggest that the empirical $I_n$ function mimics the production and decay of the

granular temperature $R$. Indeed, there is a strong qualitative similarity between the calculated $I_n$ and
$R$ functions. When the two dimensionless parameters $I_n/I_0$ and $R/R_0$ are plotted over time (Fig. 3d) or as
a function of the calculated velocity (Fig. 3e) there is both a strong qualitative and quantitative agreement.
Because $I_n$ is a pure function of velocity (for a constant height), the calculated friction $\mu(I)$ exhibits
no change during the accelerative and decelerative phases of the flow: it ascends and descends on the
same path (Fig. 3f). In contrast, because $R$ is a result of a production/decay equation it exhibits a
hysteresis (the friction does not follow the same path in the accelerative/decelerative phases of the flow).

Hysteresis effects have been observed in experiments with granular materials (Platzer et al., 2004;

Bartelt et al., 2007) and grain flows of snow (Platzer et al., 2007, Bartelt et al., 2015). They indicate a
process-dependent flow rheology that cannot be described by rheologies with constant flow parameters
(e.g., $\mu(V)$). They suggest that the friction must change as the state of the flow changes, for example as
the grain flow continuum changes velocity. The correspondence between $\mu(I)$ and $\mu(R)$ models
underscores the importance of embracing randomness and temporal evolution in the modeling of granular
flows.

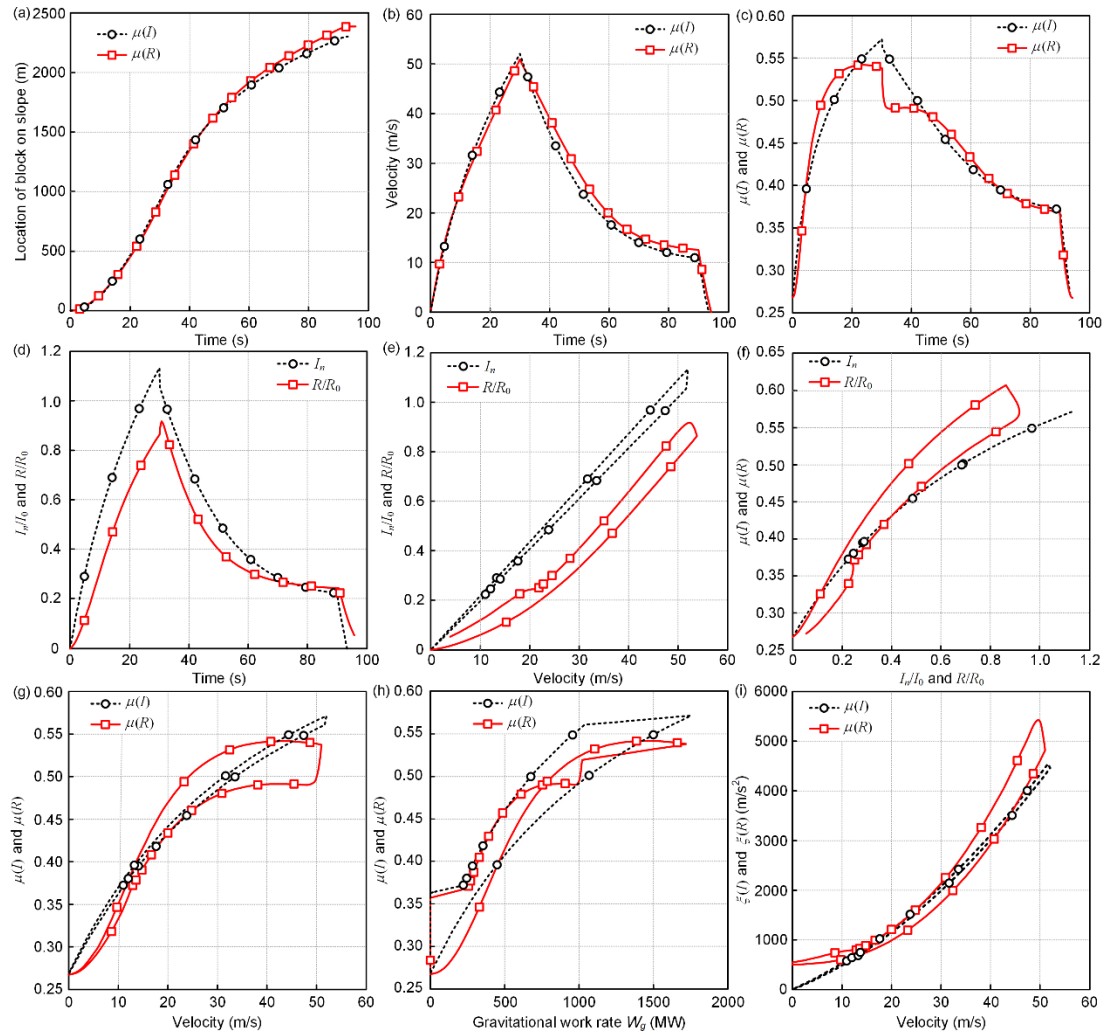

**Figure 3.** Comparison between the $\mu(I)$ vs $\mu(R)$ rheologies. (a)-(c) show the calculated location of center-of-mass, velocity and friction of the two rheologies. (d)-(e) Comparison between $I_n/I_0$ and $R/R_0$ over time and flow velocity. (f) Calculated friction $\mu(I)$ vs $\mu(R)$ as a function of $I_n/I_0$ and $R/R_0$. (g)-(h) Calculated $\mu(I)$ vs $\mu(R)$ as a function of the velocity and gravitational work rate. (i) Comparison between $\xi(I)$ (Eq. 7) and $\xi(R)$.

Both $\mu(I)$ and $\mu(R)$ rheologies exhibit hysteresis in terms of velocity (Fig. 3g) or gravitational work rate (Fig. 3h). Although the $\mu(I)$ friction expressed in terms of $I_n/I_0$ exhibits no hysteresis (Fig. 3f), the $\mu(I)$ rheology in terms of velocity and gravitational work rate does. However, this dependency is much more prominent in the $\mu(R)$-type rheologies because it is governed by two processes-both the production of fluctuation energy and its eventual decay. The $\mu(I)$ approach models the net production, always assuming that the two are in balance. During slope transitions, or other flow states in which production and decay are out-of-balance, this might not be the appropriate description. This is why the

most apparent differences between $\mu(I)$ and $\mu(R)$ arise during slope transitions. Despite these
differences, however, there is a strong correlation between $\mu(I)$ and $\mu(R)$. For example, when we
depict the calculate $\xi(I)$ and $\xi(R)$ function in terms of velocity there is almost a one-to-one agreement
in the numerical values (Fig. 3i). The only significant difference is that the $\mu(I)$ rheology predicts an
infinite friction ($\xi(I)=0$) at the velocity of zero, whereas the $\mu(R)$ approach predicts some finite value
(in this case when $R=0$, $\xi(R)=\xi_0$).

**3.2 Rheology comparison using real case studies**

**(1) Piz Cengalo rock-ice avalanche**

We apply the $\mu(I)$, $\mu(V)$, and $\mu(R)$ rheologies to calculate the dynamics of the Piz Cengalo rock-ice
avalanche and the Vallée de la Sionne snow avalanche (Sovilla et al., 2018). Modeling parameters and
results for the Piz Cengalo avalanche are presented in Fig. 4. The $\mu(R)$ parameters are empirical values,
which arise from numerous practical experiences and have been widely used in rock-ice avalanche
research (Munch et al., 2024; Zhuang et al., 2024). The input parameters ($\mu_{s,rock}$, $\mu_{s,ice}$, $\xi_{s,rock}$, $\xi_{s,ice}$)
represent the frictional parameters for a dense, granular packing of rock-ice mixture. Here, the Columb
and turbulent friction coefficients $<\mu_s(R), \xi(R)>$ are both functions of the fluctuation energy. In the
$\mu(I)$ rheology, $I_0=0.3$ is a typical value from Pouliquen & Forterre (2002), Forterre & Pouliquen (2003),
and Jop et al. (2006), $d$=1.0 m and $\mu_2$=tan(40°)=0.839 arise from field investigations of particle size and
deposit distribution. The $\mu_s$ value and parameters in the $\mu(V)$ rheology are determined from inversion
analysis that the calculated avalanche runout matches the actual condition. For ease of comparison, the
same Coulomb friction coefficients are applied in the $\mu(I)$ and $\mu(V)$ rheologies. The sensitivity
analysis of parameters in the $\mu(I)$ and $\mu(V)$ rheologies are well presented (Iannacone et al., 2013;
Argentin et al., 2022; Zhao et al., 2024; Zhuang et al., 2023b) and are not performed here.

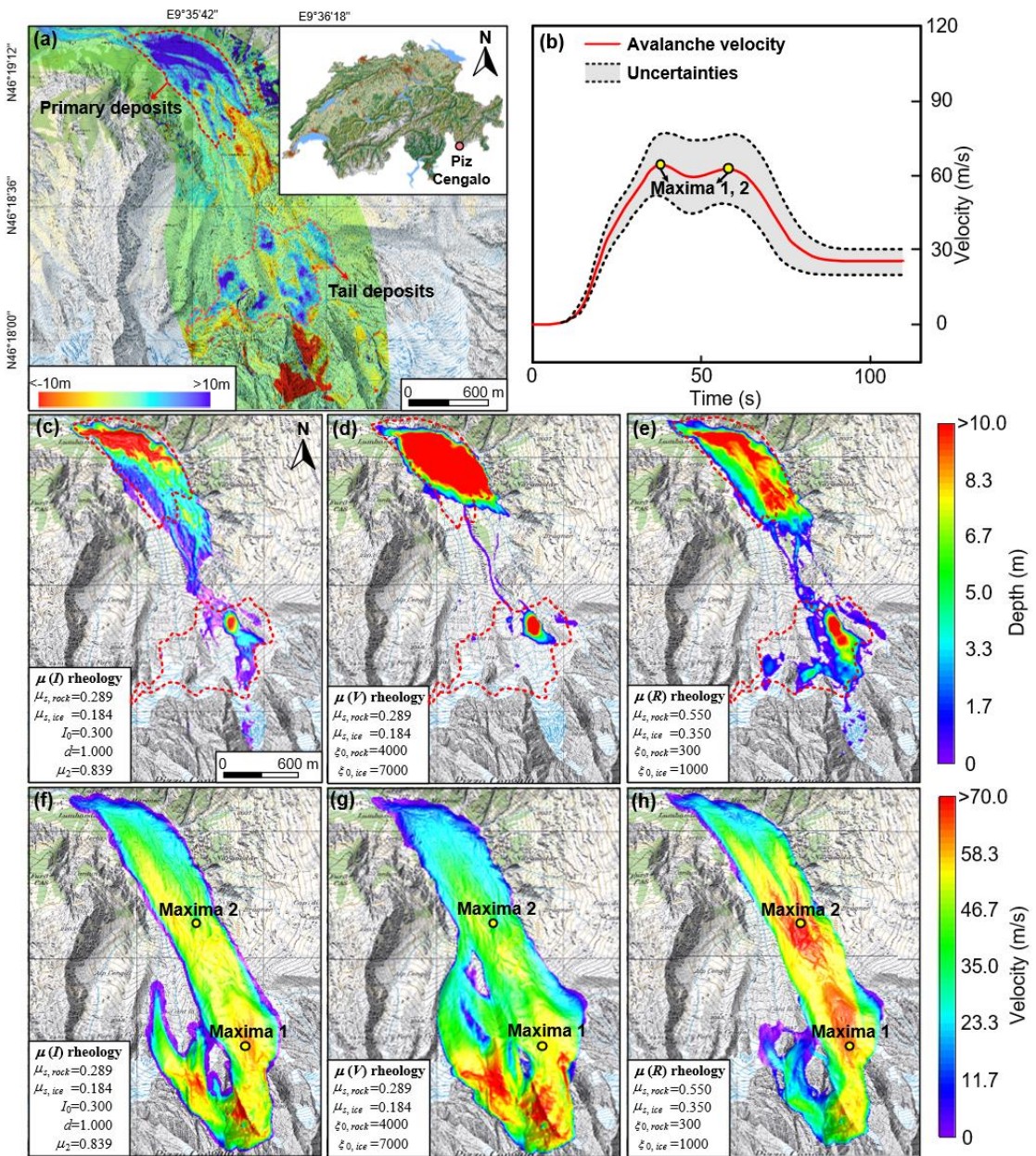


**Figure 4.** Rheology comparation with the Piz Cengalo rock-ice avalanche. (a) Deposit structure arises

from the laser scans. The grid represents the longitude and latitude of the study area. (b) Seismic signal

analysis of the avalanche velocity, derived by Walter et al. 2020. (c)-(e) Modeled avalanche deposits with

different rheologies. (4) Modeled avalanche velocity with different rheologies. Two maxima represent

the locations derived by seismic signal analysis.

Modeling results of all three rheologies exhibit satisfactory runout distance, but there are deviations

in the calculated deposit structure and avalanche velocity. Laser scans indicate two deposit areas of the

Piz Cengalo avalanche (Fig. 4a): a primary deposit area of $\sim 2 \times 10^5$ m$^2$ at the mountain toe (1350-1450

m a.s.l.) and tail deposits spread on the steep slope (2000 m-2250 m a.s.l.). Both $\mu(I)$ and $\mu(V)$ models

make a deposit anomaly at the mountain toe (Fig. 4 c and d), exceeding the measurements considerably.
Very few deposits remained on the steep slope, resulting in significantly smaller accumulation area and
thickness compared to the actual condition. Conversely, modeling deposits of the $\mu(R)$ model exhibits
a reasonable deposit structure, whether in the primary deposit area or on the steep slope (Fig. 4e). To
align the calculated avalanche runout with the actual condition, small Columb friction $\mu_s$, which is
dominant when the avalanche comes close to stopping, is applied in the $\mu(I)$ and $\mu(V)$ models. This
modification dictates the final runout accumulation, leading to deposits primarily concentrated on areas
with gentle slopes, while leaving smaller deposits on steeper inclines. According to the seismic signal
analysis (Fig. 4b, Walter et al., 2020), the Piz Cengalo avalanche has a duration of ~100 s and a maximum
velocity of 64 m/s. There are two avalanche velocity maxima: the first reaches when the avalanche leaves
the steep glacier portion, and the second occurs behind the steep terrain step in the central runout area.
The mean velocity between the two maxima is 40-60 m/s. The analysis comparing modeled avalanche
velocities and seismic signals indicates that the $\mu(R)$ rheology outperforms others in terms of peak
values and velocity evolution, as shown in Fig. 4h. Seismic signal analysis, representing the average
velocity of the mass center, explains why a slightly higher peak velocity is observed in the modeling
results. In contrast, the $\mu(I)$ and $\mu(V)$ rheologies display higher velocities downstream from the
source area but show reduced velocities in the transition and deposition areas, deviating from actual
conditions as depicted in Figs. 4f and 4g. The small Columb friction $\mu_s$ and high $\xi_0$ value impart the
avalanche with high mobility in the initial stage. This result is also visualized in the modeled deposit
distribution that very few materials are deposited on the steep slope.

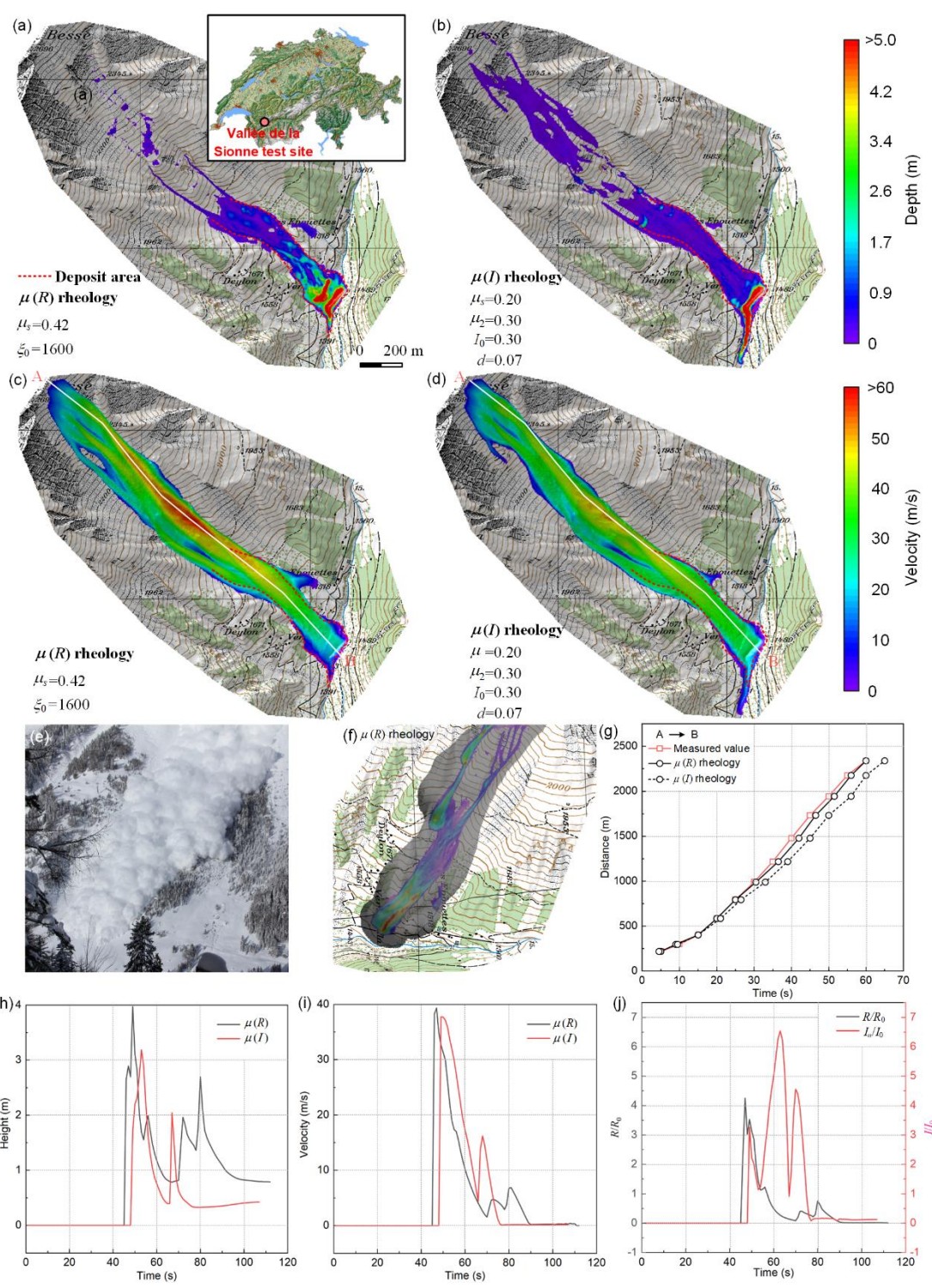


**Figure 5.** Modeling results of the Vallée de la Sionne snow avalanche (#20163017). (a)-(d) show the
simulated avalanche deposits and velocity with the two rheologies. The grid represents the longitude and
latitude of the study area. (e)-(f) show the comparison between recorded videos and modeling results of
the $\mu(R)$ rheology. (g) Comparison between measured avalanche evolution with modeling results. The
profile AB is presented in Fig. 5c-d (h)-(i) The simulated height and velocity of the mass centre with the
two rheologies. (j) Comparison between $R/R_0$ and $I_n/I_0$.


**(2) Vallée de la Sionne snow avalanche (#20163017)**
For the analyzed snow avalanche, the modeling parameters were calibrated to align the simulated
avalanche evolution and velocity with the measured values. The progression of the avalanche front was
recorded at fixed time intervals of 5 seconds, providing a basis for comparison. The modeling parameters
and results for the $\mu(I)$ and $\mu(R)$ rheologies are illustrated in Fig. 5.
Both rheological models capture the avalanche's evolution and velocity satisfactorily, though the
rheology underestimates the timing by approximately 5 seconds compared to the actual conditions (Fig.
5c, d, and g). Yet, profound differences $\mu(I)$ emerge when examining the simulated runout distance and
deposit structure. In the $\mu(R)$ rheology, the avalanche achieves a runout distance of approximately 2500
meters. The deposits are concentrated at the mountain's toe, where the slope transitions to a gentler incline,
closely mirroring field observations (Fig. 5a, e, and f).
In contrast, the $\mu(I)$ rheology exhibits significantly different behavior. The avalanche does not stop
at the mountain's toe but continues moving into the valley, showing excessive mobility (Fig. 5b). The
sliding mass bulks unnaturally in the valley, and the deposit depth greatly exceeds observed conditions.
This divergence arises from the Coulomb friction coefficient $\mu_s$ used in the $\mu(I)$ rheology. To match
the measured velocity, a smaller $\mu_s$ value was applied, resulting in an extended runout and deposition
in the flatter terrain of the valley.
Further insight emerges when contrasting $R/R_0$ with and $I_n/I_0$, as shown in Fig. 5j. The scaling
factors $R_0$ and $I_0$ encapsulate the influence of sliding materials. While $R_0 = 2$ kJ/m$^3$ represents a
typical value for snow avalanches (Buser & Bartelt, 2015), $I_0$ is derived from laboratory experiments
using glass beads (Forterre & Pouliquen, 2003; Jop et al., 2006). This disparity in scaling reflects the
intrinsic differences in material behavior and introduces a subtle, yet significant, divergence in
rheological interpretation.
Through this analysis, we observe that the $\mu(R)$ rheology, with its non-steady production and
dissipation of fluctuation energy, achieves a more faithful reproduction of both the avalanche's dynamics
and deposition patterns, underscoring the nuanced interplay of microscopic and macroscopic principles
in granular flow systems.
**4. Discussion and Implications**
With this contribution, we strengthen the theoretical foundation of the $\mu(I)$ rheology. It has an
equivalence with the Voellmy-type grain flow rheologies, which are composed of a Coulomb stopping
friction and a turbulent friction that controls the flow velocity. Compared with the classic $\mu(V)$ rheology
of constant friction parameters, an advantage of the $\mu(I)$ rheology is to define the turbulent friction
parameter $\xi(I)$ as a function of flowing velocity and height (using inertial number $I_n$). This modification
incorporates the shear-thinning behavior (Hu et al., 2022) and the impact of volume (where increased
normal stress results in a reduced friction coefficient, see Heim, 1932; Wang et al., 2018), capturing key
characteristics of these phenomena. With the help of grain flow theory (Haff, 1983, Jenkins & Savage,
1983; Buser & Bartelt, 2009), we find the contribution of $I_n$ attributes to its empirical representation of
the granular temperature/fluctuation energy $R$. However, the inertial number $I_n$ is just a function of
flowing velocity, assuming the production and decay of the fluctuation energy are in balance. The $\mu(I)$
rheology, therefore, exhibits no change during the acceleration and deceleration process, leading to the
deviation of calculated velocity for real case studies.
Though the $\mu(I)$ rheology demonstrates an improvement over the classic $\mu(V)$ rheology, it has a
critical flaw in ignoring the contribution of fluctuation energy to the Coulomb friction coefficient $\mu_s$. In
the $\mu(I)$ rheology, the constant $\mu_s$ value makes the sliding mass stop on a single slope angle
(arctan($\mu_s$)). Consequently, the modeled deposits of the Piz Cengalo avalanche and Vallée de la Sionne
snow avalanche concentrate at the mountain toe, with very few materials deposit on the slope.
Considering that avalanche deposits in real-world scenarios often cover a broad area with varying
thicknesses, using a constant $\mu_s$ value is unlikely to yield an accurate representation of the deposit
structure.
A significant challenge in landslide risk assessment is to establish reliable numerical parameters,
highlighting a limitation in both the $\mu(I)$ and classic $\mu(V)$ rheologies: the reliance on input parameters
derived from inversion analysis (Zhao et al., 2024). Although the $\mu(I)$ rheology is based on
experimental data, relevant experiments are limited, and the test materials used are predominantly glass
beads (Foterre & Pouliquen, 2003; Jop et al., 2006). To date, no large-scale experiments have been
conducted on geophysical mass flows, to our knowledge. Considering the substantial differences in
properties among materials in the flowing mass, such as rock, ice, snow, and water, it proves highly
challenging to accurately characterize avalanche motion using a uniform surrogate material with different
properties, such as glass. Additionally, the dynamics of avalanches are greatly influenced by the flow
regime and topography, indicating that avalanches composed of the same material can display varied
runout lengths and deposit patterns under different conditions.
This phenomenon further complicates the task of selecting appropriate model parameters. In this
study, to achieve a satisfactory runout of the Piz Cengalo avalanche and a reasonable velocity of the
Vallée de la Sionne snow avalanche, small $\mu_s$ values arise from inversion analysis are applied for the
calculation of $\mu(I)$ and $\mu(V)$ models. We admit that model parameters can be calibrated such that
realistic runout or velocity are obtained, but these site-specifically calibrated parameters limit the
engineering application of the model, particularly when conducting risk assessments of potential
avalanches. The existing $\mu(R)$ model offers a possible solution (Christen et al., 2010; Bartelt et al., 2011;
Zhuang et al., 2023c). By defining the Coulomb stopping friction and turbulent friction parameters as
functions of fluctuation energy, we can characterize the effects of flow regime and topography changes
on the friction of landslides (Preuth et al., 2010). Using a group of empirical parameters, which represent
the material properties of rock, ice and snow, realistic deposit structure and velocity evolution can be
obtained. Because $R$ represents the energy associated with random particle motions, it introduces an
element of stochasticity into avalanche modelling. Clearly, it is impossible to precisely determine the
position of every individual particle in an avalanche, contrary to what Discrete Element Modeling (DEM)
might imply. Nonetheless, the behavior of the granular ensemble seems to be directed by a
production/decay equation, which, even when estimated approximately, can impart a discernible
trajectory to the avalanche process and deposition dynamic, thereby enhancing the predictive accuracy
of numerical models.

Further case studies on various types of geophysical mass flows, such as rock avalanches, ice

avalanches, and snow avalanches, will help quantify the modeling parameters of $\mu(R)$ rheology
(production and decay of fluctuation energy) with less uncertainty. The remaining challenge is to
formulate a comprehensive rheology that incorporates the critical physical processes involved in mass
flows, including water lubrication, fluidization, sliding materials, and ground roughness.
**5. Conclusion**
In this paper, we describe the equivalence and difference between three widely-used rheologies to model
geophysical mass flows: (1) the classic Voellmy rheology, (2) $\mu(I)$ rheology and (3) $\mu(R)$ rheology.
The $\mu(I)$ rheology can be reformulated as Voellmy-type, which is composed of a Coulomb and a
turbulent friction term. Different from the classic Voellmy rheology (constant $\xi$ value), $\mu(I)$ rheology
involves a velocity-dependent $\xi$ parameter, modeling a shear-thinning behavior. It utilizes a
dimensionless inertial number $I_n$ to minic contributions of fluctuation energy to the runout behavior of
mass flows, building an equivalence with the $\mu(R)$ rheology. Though both $\mu(I)$ and $\mu(R)$ models
indicate that friction is a process, changing in time and space, the $\mu(I)$ rheology assumes the production
and decay of fluctuation energy are in balance, exhibiting the same friction behavior during the
accelerative and depositional phases. More importantly, a critical flaw of the $\mu(I)$ rheology is
suggesting a constant Colomb friction, ignoring the impacts of fluctuation energy on the Colomb
stopping friction. Modeled avalanche deposits of the Piz Cengalo rock-ice avalanche and the Vallée de
la Sionne snow avalanche are both concentrated in areas with gentle slopes. The existing $\mu(R)$ rheology
makes up for the shortcomings, exhibiting good performance in predicting the deposit patterns of
geophysical mass flows. These insights have practical implications for improving geophysical flow
models, offering a more comprehensive understanding of flow behavior and its dependence on factors
such as velocity, terrain features, and material properties. As we continue to refine our models, we move
closer to more accurate assessments and mitigation of geophysical hazards.
**Data availability**
No data sets were used in this article.
**Author contribution**
Yu Zhuang did the numerical work and wrote the manuscript with contributions from all co-authors.
Perry Bartelt designed the work, did the calculation and wrote the manuscript. Brian W. McArdell edited
the manuscript.
**Competing interests**

The authors declare that they have no known competing financial interests or personal relationships that

could have appeared to influence the work reported in this paper.

**Acknowledgments**

This study is supported by the RAMMS project and the Fundamental Research Funds for the Central

Universities.

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
