# Peer review of "Comparative Analysis of $\mu(I)$ and Voellmy-Type Grain Flow Rheologies in"

_Natural Hazards and Earth System Sciences, 2024_

## Author Response (AR1)

**Reviewer 1:**

This manuscript, based on theoretical derivation and practical cases, comparatively analyzes the  $\mu(I)$  and Voellmy-type rheological relationships in geophysical mass flows. The study has positive value for modeling granular flows. Specific suggestions are as follows:

(1) Please clearly specify the information about the case study in the abstract.

**Response:** Following your comment, we provided the information about the case study in the abstract (Line 13).

(2) Provide more information about Figure 4, including the latitude and longitude grid and the location of the event on a larger map, so that readers can easily identify where the event occurred.

**Response:** Many thanks for your comment. The grid in the image represents the longitude and latitude of the study area. Following your comment, we further provided a larger map to show the location of the event (Fig. 4a).

(3) In the discussion, it is recommended to add a description and analysis of the limitations of the methods and results of this manuscript, as well as future work prospects.

**Response:** Many thanks for your comment. One of our goals is to talk about the advantages and limitations of the classic Voellmy rheology and  $\mu(I)$  rheology in calculating the mass flow movement. We have made a discussion and analysis in "Discussion". We talked about the limitations of the Voellmy  $\mu(V)$  rheology (Lines 321-325, 340-344, 351-357), the limitations of the  $\mu(I)$  rheology (Lines 328-357). Following your valuable comment, we further made a better analysis about future work prospects in the revised manuscript (Lines 369-373).

(4) Add a conclusion section. In summary, major revisions are recommended."

**Response:** Following your comment, we added a conclusion section in the revised manuscript (Lines 374-392).

**Reviewer 2:**

This study compares the  $\mu(I)$  rheology and Voellmy-type grain flow rheologies, revealing their equivalences and differences through theoretical and case study analyses. The results not only deepen our understanding of the dynamics of geophysical mass flows but also highlight practical engineering challenges, providing theoretical support for future natural hazard risk assessment and mitigation. The paper is well written and structured. However, in this reviewer's opinion it suffers some minor issues that need to be addressed before being considered for publication in NHESS.

Specific comments are following below.

(1) Could you please provide specific quantitative results or key data in the abstract to give readers immediate insight into the magnitude of differences or similarities found between the  $\mu(I)$  and Voellmy-type grain flow rheologies?

**Response:** Many thanks for your valuable comment. We can provide specific quantitative results in the abstract to make readers a better understand (Lines 22-27).

(2) More detailed explanation about the key terms should be given, such as "granular

temperature/fluctuation energy" and "inertial number".

**Response:** The fluctuation energy (granular temperature) is a term in the field of thermodynamics, which is defined from the velocity fluctuation (Campbell, 2006). The inertial number is a dimensionless number that quantifies the significance of dynamic effects on the flow of a granular material. It measures the ratio of inertial forces of grains to imposed forces. For a granular flow, the average inertial number can be written as  $I_n = \frac{5}{2h} \frac{vd}{\sqrt{g_z h}}$  (Eq. 5 in the manuscript, GDR MIDI, 2004). Following your advice, we provide explanations about these terms in the revised manuscript (Lines 71-74, 92-93).

Campbell, C. S.: Granular material flows-An overview, Powder Technology, 162, 208-229, 2006. GDR, MiD.: On dense granular flows, The European Physical Journal E, 14, 341-365, 2004.

(3) Authors should expand the discussion on the calibration and validation of the models used in simulations. Specifically, address how the model parameters were chosen and their sensitivity to the predicted outcomes.

**Response:** Many thanks for your comment. Actually, the Voellmy  $\mu(V)$ ,  $\mu(I)$  and  $\mu(R)$  rheologies have been widely applied in the gravitational mass flow calculations and the sensitivity analysis of rheological parameters has been conducted. All three rheological models are validated in many previous literatures. In this study, we didn't develop the model but compare the three widely-used rheologies, trying to bolster the theoretical foundation of mass flow modeling. Therefore, following your comment and making the paper concise, we added some related references in the revised manuscript (Lines 249-251).

For the parameter selection, the rheological parameters for the  $\mu(R)$  rheology are empirical values, arising from numerous case studies. The input parameters ( $\mu_{s,rock}$ ,  $\mu_{s,ice}$ ,  $\xi_{s,rock}$ ,  $\xi_{s,ice}$ ) represent the frictional parameters for a dense, granular packing of rock-ice mixture (Munch et al., 2024; Zhuang et al., 2024). For the Voellmy  $\mu(V)$  rheology, the parameters are determined from back analysis until the modeling avalanche runout matches the actual condition. For the  $\mu(I)$  rheology,  $I_0=0.3$  is a typical value from Pouliquen & Forterre (2002), Forterre & Pouliquen (2003), and Jop et al. (2006), d=1.0 m and  $\mu_2=\tan(40^\circ)=0.839$  arise from field investigations of particle size and deposit distribution. The  $\mu_s$  value arises from the back analysis, the same as the Voellmy  $\mu(V)$  rheology. Following your comment, we provided the explanation in the revised manuscript (Lines 240-249).

- Forterre, Y., and Pouliquen, O.: Long-surface-wave instability in dense granular flows, Journal of Fluid Mechanics, 486, 21-50, 2003.
- Jop, P., Forterre, Y., and Pouliquen, O.: A constitutive law for dense granular flows, Nature, 441(7094), 727-730, 2006.
- Munch, J., Zhuang, Y., Dash, R. K., and Bartelt, P.: Dynamic Thermomechanical Modeling of Rock-Ice Avalanches: Understanding Flow Transitions, Water Dynamics, and Uncertainties, Journal of Geophysical Research: Earth Surface, Authorea Preprints.
- Pouliquen, O., and Forterre, Y.: Friction law for dense granular flows: application to the motion of a mass down a rough inclined plane, Journal of fluid mechanics, 453, 133-151, 2002.
- Zhuang, Y., Dawadi, B., Steiner, J., Dash, R. K., Bühler, Y., Munch, J., and Bartelt, P.: An earthquake-triggered avalanche in Nepal in 2015 was exacerbated by climate variability and snowfall anomalies. Communications Earth & Environment, 5, 465, 2024.

(4) Discuss specific instances or types of geophysical events where the comparative analysis of these rheologies could be particularly beneficial. if possible, please provide one more case study or

hypothetical application.

**Response:** Many thanks for your comment. We added another snow avalanche case in the revised manuscript. We did calculations with the  $\mu(I)$  and  $\mu(R)$  rheologies and discuss the similarity and differences of the modeling results (Lines 291-317).

---

## Author Response (AR2)

Dear Yu Zhuang,

We are pleased to inform you that your following manuscript was accepted for final publication in NHESS:

nhess-2024-87
Title: Comparative Analysis of μ(I) and Voellmy-Type Grain Flow Rheologies in Geophysical Mass Flows: Insights from Theoretical and Real Case Studies
Author(s): Yu Zhuang et al.
MS type: Research article
Iteration: Major revision

Presently, your manuscript is being transferred to the Copernicus Publications Production Office for typesetting and publication. To proceed, please log in with your Copernicus Office user ID 662587 to upload all files that are required for production no later than 20 Mar 2025 at: https://editor.copernicus.org/NHESS/production_file_upload/nhess-2024-87

For further information on files and formats, we kindly refer you to the submission guidelines at: https://www.natural-hazards-and-earth-system-sciences.net/for_authors/submit_your_manuscript.html

In your manuscript, please use full first names for all authors. Although references are still based on initials, we will use full first names on the title page of your paper.

Please ensure that the reproduction rights for all figures have already been secured and that maps and aerials include the required copyright statements or credits as requested by the providers.

Before file upload, please consider submitting data sets, model code, or video supplements to reliable repositories, receive DOIs, and cite these assets in your manuscript including entries in the reference list.

To promote your work, please provide a 500-character short summary during production file upload and consider producing a short video abstract. Upload your video abstract to an appropriate video portal, provide the link/DOI during production file upload, and we will embed your video in your article's web page.

You are invited to monitor the processing of your manuscript via your MS overview at: https://editor.copernicus.org/NHESS/my_manuscript_overview

Thank you very much in advance for your cooperation. In case any questions arise, please do not hesitate to contact me.

Kind regards,

The editorial support team
Copernicus Publications
editorial@copernicus.org

**Response:** We appreciate the efforts made by the editor and two reviewers to improve the quality of the paper.